# New-Onset Acute Kidney Disease Post COVID-19 Vaccination

**DOI:** 10.3390/vaccines10050742

**Published:** 2022-05-09

**Authors:** Yebei Li, Meiying Rao, Gaosi Xu

**Affiliations:** 1Department of Nephrology, The Second Affiliated Hospital of Nanchang University, No. 1, Minde Road, Donghu District, Nanchang 330006, China; liuli666@email.ncu.edu.cn; 2Department of Blood Transfusion, The Second Affiliated Hospital of Nanchang University, No. 1, Minde Road, Donghu District, Nanchang 330006, China; 411439818385@email.ncu.edu.cn

**Keywords:** acute kidney disease, acute kidney injury, COVID-19, SARS-CoV-2, vaccination

## Abstract

The coronavirus disease 2019 (COVID-19) pandemic, caused by the severe acute respiratory syndrome coronavirus 2 (SARS-CoV-2), has caused an exceptional setback to the global economy and health. Vaccination is one of the most effective interventions to markedly decrease severe illness and death from COVID-19. In recent years, there have been increasingly more reports of new acute kidney injury (AKI) after COVID-19 vaccination. Podocyte injury, IgA nephropathy, vasculitis, tubulointerstitial injury, and thrombotic microangiopathy appear to be the main pathological phenotypes. Nonetheless, whether the link between the COVID-19 vaccine and acute kidney disease (AKD) is causal or coincidental remains to be verified. Here, we generalize some hypotheses for the emergence of AKD and its pathogenesis in response to certain COVID-19 vaccines. In fact, the enormous benefits of mass vaccination against COVID-19 in preventing COVID-19 morbidity and mortality cannot be denied. The purpose of this review is to assist in the clinical assessment and management of AKD following COVID-19 vaccination.

## 1. Introduction

With the ongoing coronavirus disease 2019 (COVID-19) pandemic and the emergence of new variants of severe acute respiratory syndrome coronavirus type 2 (SARS-CoV-2), the rapid development of effective and safe preventive vaccines is urgently required to control disease outbreaks [1,2]. Over the past 2 years, hundreds of COVID-19 vaccine candidates have been developed, tested, and finally rolled out, including protein-based vaccines (Novavax), inactivated vaccines (Sinovac Life Science), viral vector vaccines (Janssen, Oxford-AstraZeneca), and mRNA vaccines (Pfizer/BioNtech, Moderna, CureVac) (Figure 1) [2,3]. Among them, mRNA-based drugs are new but not unknown [4]. mRNA vaccines deliver transgenic mRNA through lipid nanoparticles, which act as carriers. Once injected, the mRNA is translated into the target protein in vivo, resulting in a strong immune response, and a 2-dose regimen confers 95% protection against COVID-19 [5]. To date, large phase III and IV trials have found these vaccines to have a good safety profile, with few serious reactions [3,6,7,8,9]. Common short-term adverse events include local injection site reactions, fever, fatigue, generalized pain, and headache [6,10].

However, since mass vaccination, there have been a few case reports of acute kidney injury (AKI), acute kidney disease (AKD), proteinuria, edema, gross hematuria, and other renal side effects requiring hospitalization after COVID-19 vaccinations [11]. Serum creatinine (Scr) levels and proteinuria recovered within 3 months of treatment in most patients. The vast majority of cases occurred after mRNA vaccine and adenoviral vector injection, and a few cases of glomerulonephritis associated with inactivated virus vaccines have also been reported.

In this review, we summarize the clinical features of AKD after vaccination, and elaborate on the possible mechanisms of AKD after COVID-19 vaccination.

## 2. Clinical Characteristics of Patients

In this review, we performed a literature search through the electronic database, including Medline/PubMed, EMBASE, and Web of Science (WOS) before February 2022, using the following keywords: (“acute kidney injury” OR “acute kidney disease” OR “glomerulonephritis”) AND (“COVID-19” OR 2019-nCoV” OR “novel corona virus” OR “SARS-CoV-2” OR “coronavirus”) AND (“vaccine” OR “vaccination”).

According to the Kidney Disease: Improving Global Outcomes (KDIGO) criteria, AKI stage 1 is defined as a 0.3 mg/dL increase in Scr within 48 h or a 1.5 to 1.9 times increase in Scr from baseline within 7 days, AKI stage 2 is defined as a 2 to 2.9 times increase in Scr within 7 days from baseline, and AKI stage 3 is defined as a 3 times or more increase in Scr within 7 days from baseline or initiation of renal replacement therapy (RRT) [12,13]. AKD is defined as a condition of acute or subacute damage and/or loss of renal function between 7 and 90 days after exposure to an AKI initiating event [14]. AKD lasting more than 90 days is considered to be chronic kidney disease (CKD) [14]. The baseline Scr was determined by a median Scr within 8 to 365 days prior to admission. Because of inaccurate and missing urine output records in electronic health records, we did not use the urine output criteria in the KDIGO guidelines to define AKI [15].

As of 28 February 2022, we found a total of 38 published articles (53 cases) on AKI after SARS-CoV-2 vaccination (Table 1) [11,16,17,18,19,20,21,22,23,24,25,26,27,28,29,30,31,32,33,34,35,36,37,38,39,40,41,42,43,44,45,46,47,48,49,50,51,52]. Only one case clearly identified an increase in Scr and a resolve within 7 days after vaccination [30]. Scr returned to baseline within 90 days in 37 patients, and no outcomes were recorded in 6 patients. Seven patients did not respond.

There were 53 cases in total, including 47 (89%) cases of new kidney involvements and 6 (11%) cases of relapse. Minimal change disease (MCD) was the most common pathology (13 (25%): 11 new, 2 relapse), followed by IgA nephropathy (IgAN) (11 (21%): 9 new, 2 relapse), acquired thrombotic thrombocytopenic purpura (aTTP) (8 (15%): all new), and anti-neutrophil cytoplasmic autoantibodies (ANCA) vasculitis (8 (15%): all new). Other diagnosed diseases included acute interstitial nephritis (AIN) in four new cases, membranous nephropathy (MN) in four cases (two new, two relapse), anti-glomerular basement membrane (anti-GBM) nephritis in three cases (all new), ANCA-negative granulomatous vasculitis in one new case, and leukocytoclastic vasculitis in one new case. The mean age was 58 years (standard deviation 18), and 64% were male.

The Pfizer-BioNTech (mRNA) vaccine was the most common vaccine used (27/53 patients, 51%), followed by Moderna (mRNA) (16/53 patients, 30%), AstraZeneca (Adenovirus vector) (5/53 patients, 9%), Sinovac Life Science (Inactivated virus) (3/53 patients, 6%), and Janssen (Adenovirus vector) (3/53 patients, 6%). Of the 53 patients, 26 patients (49%) had an increase in Scr after the first dose, 25 patients (47%) after the second dose, and 2 patients (4%) after both doses.

## 3. Clinical Characteristics and Follow-Up of Patients by Disease

### 3.1. Minimal Change Disease (MCD)

Of all the cases reported in the literature, MCD was the most common pathological type of AKD after COVID-19 vaccinations, with a total of 13 cases (11 new and 2 relapse). Six patients had acute renal tubular and/or interstitial injury. Ten patients developed AKD after the first vaccination, and three developed AKD after the second dose. Edema is the most common symptom. All patients received steroids therapy, and one patient also underwent hemodialysis [16]. A patient with relapsed MCD who did not respond to high-dose steroids received rituximab (RTX), to which the patient responded [11]. In total, 11 patients achieved complete remission (CR) or partial remission (PR) within 3 months of treatment, 1 had no response [20], and 1 had no follow-up records [21].

### 3.2. IgA Nephropathy (IgAN)

Eleven cases of IgAN (nine new onset, two relapse) were reported in the literature, of which only one patient could be identified with AKI without AKD [30]. One case was complicated with AIN [11]. Gross hematuria was the most common presentation. Six patients (four new, two relapse) received conservative treatment, four achieved CR, one did not achieve normal Scr, and one had no follow-up records. The other five new-onset patients were treated with steroids, three patients responded to treatment, and two patients had no follow-up records.

### 3.3. Membranous Nephropathy (MN)

MN developed in four patients (two new and two relapse), and three were serum anti-phospholipase A2 receptor (PLA2R) antibody positive. All patients had symptoms of edema. One new-onset patient received conservative treatment with no remission within 60 days [26]. The other new case developed MN and AKD after the first mRNA (Pfizer-BioNTech) vaccination, achieved partial remission after administration of renin-angiotensin system blockade (RASB), and Scr did not return to baseline levels. Edema reappeared after the second mRNA (Moderna) vaccination, and Scr remained at 1.15 mg/dL [27]. The patient was subsequently given rituximab (RTX) and achieved a PR. One relapse case responded to tacrolimus (TAC), and Scr did not return to baseline during the last follow-up [11]. The other relapsed patient had no records of treatment or follow-up [42].

### 3.4. Anti-Glomerular Basement Membrane (Anti-GBM) and Anti-Neutrophil Cytoplasmic Autoantibodies (ANCA) Vasculitis

According to reports in the literature, there have been three cases of typical new-onset anti-GBM nephritis, accompanied by gross hematuria, hypertension, anorexia, nausea, fever, and other symptoms. One case did not respond to mycophenolate and steroids, and his Scr level continued to rise [11]. Two patients received cyclophosphamide (CyC), plasma exchange (PLEX), and pulse methylprednisolone therapy, one patient did not respond and continued dialysis [32], and the other one patient had no follow-up records [29].

In the literature, eight cases were new-onset ANCA-associated vasculitis, five cases were associated with myeloperoxidase (MPO), and three cases with proteinase 3 (PR3). In addition, there was one report of ANCA-negative granulomatous vasculitis following the first dose of adenoviral vector (AstraZeneca) vaccination, and the patient’s Scr returned to normal within 2 months [39]. All 3 patients with confirmed PR3-ANCA vasculitis developed symptoms within 1 week after the second dose of mRNA vaccine and were then admitted to hospital for treatment. One case received RTX, CyC, PSL, and hemodialysis treatment, and received hemodialysis without remission. Further, 1 case received high-dose glucocorticoids, CyC, and PLEX, and achieved Scr remission within 3 weeks, and 1 case received pulse methylprednisolone therapy, RTX, CyC, and PLEX, and Scr decreased to 1.5 mg/dL within 10 weeks. From the literature, both MPO-ANCA and ANCA-negative patients with granulomatous vasculitis show improved Scr after treatment.

### 3.5. Acquired Thrombotic Thrombocytopenic Purpura (aTTP)

According to the statistics, only one of the eight patients was reported after the second vaccination [47]. The main symptoms included fever, headache, fatigue, ecchymosis on the limbs, and gastrointestinal reactions, which usually appeared 1 to 2 weeks after vaccination. Except for the one case reported by Al Rawahi et al. [51], all other cases were immediately treated with PLEX and immunosuppressive therapy, such as corticosteroids, RTX, etc., and the clinical effects were generally good. Additionally, in three cases, the first dose of the COVID-19 vaccine induced high titers of neutralizing IgG against SARS-CoV-2 [45,49,51]. ADAMTS13 (A Disintegrin And Metalloproteinase with a ThromboSpondin type 1 motif, member 13) activity levels were markedly reduced and its inhibitor’s titer was increased in some cases [45,46,47,49]. Hence, some researchers speculated that in patients without underlying diseases, de novo aTTP was associated with COVID-19 vaccination [49,51].

## 4. Inducing AKD through COVID-19 Vaccine: Hypotheses

### 4.1. Podocyte Damage

The temporal association between intramuscular vaccination and the development of MCD speculates that a cell-mediated immune response may be a trigger for podocyte injury [20,53]. All 12 patients with MCD reported in the literature were over 60 years of age, developed AKD within 2 weeks of vaccination, and steroids appeared to be effective in achieving rapid remission (Table 1). Typically, following vaccination, the vaccine’s antigens are taken up by dendritic cells and then presented to T cell receptors on naive T cells [54]. This leads to the activation of antigen-specific effector T cells, peaking 7 to 14 days after vaccination [55]. Studies have also confirmed that during viral infection, cellular immune responses can be observed within about 1 week after infection, but T cell activation can occur 2–3 days earlier [56,57]. This answers the question of whether it is reasonable for a COVID-19 vaccine to elicit a cell-mediated response 3–4 days after administration.

Although the exact pathogenesis of MCD remains unclear, podocyte damage caused by circulating factors released by activated T lymphocytes appears to be decisive (Figure 2) [58,59]. During active stages of MCD, T cell subsets are imbalanced, and circulating CD8+ suppresses the prevalence of T cells, which is exacerbated by cytokine-induced damage [60]. Compared with conventional vaccines, mRNA vaccines are expected to provoke higher antibody responses and stronger CD8+ T and CD4+ T cell reactions, including higher chemokine and cytokine production [61,62]. The resulting irregular permeability factors can alter glomerular permeability and lead to marked proteinuria and kidney injury [53].

Another hypothesis we speculate might be relevant is that type 2 helper T cells (Th2) indirectly induce tissue cell damage through hypersensitivity reactions via nucleic acid (NA) sensors. Previous study has demonstrated that T cells sensing their own NAs can trigger and amplify allergic inflammation independent of known NA sensors in innate immunity [63]. Muscle cells presenting viral mRNA-derived products on major histocompatibility complex class I are eliminated by CD8+ T cells, and self-NA released by dead muscle cells may directly induce T cell co-stimulation. This may be followed by Th2 differentiation and Th2-mediated allergic inflammation, causing podocytopathy [23]. Nevertheless, the study by Sahin et al. found that the COVID-19 mRNA vaccine elicited a cytokine response involving Th1 T cell responses [62,64].

Furthermore, SARS-CoV-2 can penetrate proximal tubular cells through ligation with angiotensin conversion enzyme 2 (ACE2) and CD147-spike protein to cause severe AKI, and can also penetrate podocytes through ligation with ACE2, resulting in podocyte dysfunction [65,66]. In addition, SARS-CoV-2 can also unbalance renin-angiotensin-aldosterone system (RAAS) activation, promoting inflammation, glomerular dysfunction, fibrosis, and vasoconstriction [66]. However, whether the vaccine is related to ACE2 and RAAS is unclear.

### 4.2. Increased Production of Anti-Neutrophil Cytoplasmic Autoantibodies (ANCAs)

Influenza and rabies vaccines based on viral mRNAs have been described to possibly lead to an increase in ANCA, contributing to the development of ANCA-associated vasculitis [67]. Moreover, it was confirmed that the ANCA response was significantly reduced after the treatment of vaccinees with ribonuclease. Scientists have found that in the context of COVID-19, a host response to viral RNA can directly cause ANCA-associated vasculitis (AAV) and an autoimmune response [68,69,70]. COVID-19 mRNA vaccination induced a stronger response of the innate immune system after the second booster compared with primary immunization [71]. The heightened innate immune response observed after the second vaccination with BNT162b2 mRNA vaccine may be an inducer of MPO-ANCA and PR3 autoantibodies [34]. Toll-like receptors (TLRs) can be expressed on leukocyte membranes and play an important role in inflammatory responses, recognizing viral antigens and promoting immune system activation. In AAV, major toll-like receptor 2 (TLR2) and toll-like receptor 9 (TLR9) activation can provoke autoimmunity [72]. Interestingly, Kumar et al. suggested that TLR2 was activated by a robust and specific immune response of immunodominant cytotoxic T-lymphocyte (CTL) to the spike glycoprotein of SARS-CoV2 (also produced by the COVID-19 vaccine) [73]. Messenger RNA vaccines could act as both antigen and adjuvant due to their intrinsic immunostimulatory properties of RNA; thus, they can be recognized by endosomal TLRs and cytosolic inflammasome components [64]. Therefore, the occurrence of AAV in the context of COVID-19 mRNA is highly relevant compared with non-mRNA vaccinations, but further experiments are required to verify the mechanism of the link between autoimmunity and a COVID-19 vaccine.

### 4.3. Vaccine-Induced Thrombotic Thrombocytopenia (VITT)

Some scholars have speculated that antiphospholipid antibodies (APLs) may be part of the cause of thrombosis after COVID-19 vaccination, by triggering the type I interferon response associated with APLs’ production [74,75]. It binds directly to platelets by inhibiting the anticoagulant pathway of protein C, triggers the coagulation cascade, and appears to be associated with abnormal activation of immune responses involving the complement cascade [74]. Thrombocytopenia and platelet activation have been reported following the administration of adenoviral gene transfer vectors [76]. Thrombocytopenia also occurred after treatment with some anti-sense oligonucleotides [77]. Based on the above background, another hypothesis speculates that the activation of platelets by adenovirus-platelet-leukocyte complexes, mediated by von Willebrand factor (VWF) and P-selectin, may lead to accelerated clearance of platelets in the liver [75,78].

However, the virus in viral vector vaccines is replication-incomparable and the circulating virus disappears 7–14 days after vaccination, so the viral localization to the central nervous system and digestive system causing thrombosis is unlikely [79]. In addition, Greinacher et al. suggested that the rare occurrence of VITT was mediated by platelet factor 4 (PF4)-dependent platelet-activating antibodies, which in turn stimulate platelets via their Fcγ receptors [80,81]. Immune complexes containing PF4 can be recognized by C1q, which binds to the Fc portion of IgG molecules. This results in C3 activation, expansion of the complement response, and production of downstream proinflammatory mediators and effectors, ultimately leading to enhanced thrombus inflammation.

### 4.4. Direct Induction of Myositis

A previous case reported that a patient who presented with profound left upper arm pain after COVID-19 mRNA vaccination had an increased serum creatine kinase concentration, indicating skeletal muscle damage and inflammation (myositis) [82]. There is also evidence of renal biopsies from post-vaccination patients showing massive rhabdomyolysis-induced myoglobin casting, which may contribute to worsening renal function [34].

## 5. Discussion

This review has limitations. First, most of the literature is from reported single-case studies, and we cannot infer a causal relationship between vaccines and AKD. There may be potential undetected confounding factors. Second, there may be many unreported cases that may not represent the true incidence of AKD. Third, the mechanism is unproven, with only a combination of hypotheses derived from case reports and literature.

## 6. Conclusions

Despite these reported cases of AKD, the protective role of COVID-19 vaccination far outweighs any risks identified so far. In conclusion, the occurrence of AKD is relatively rare following COVID-19 vaccination. If some symptoms, such as hematuria, foamy urine, and edema, can be detected in an early phase, patients will benefit from a timely treatment of the primary disease, such as steroids. Further research is warranted to better understand the causes and mechanisms of AKD after COVID-19 vaccination.

## Figures and Tables

**Figure 1 vaccines-10-00742-f001:**
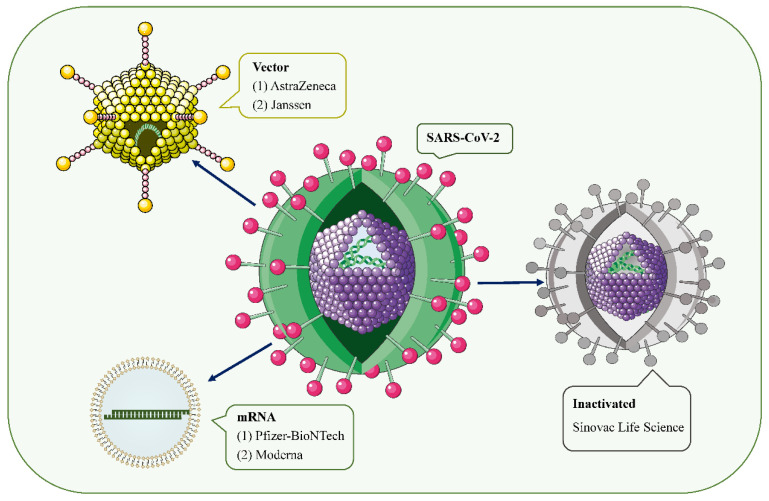
SARS-CoV-2 and the main types of vaccines that may trigger AKD. SARS-CoV-2 is a positive-sense single-stranded RNA virus with a lipid bilayer consisting of the spike S protein and membrane and envelope proteins. mRNA vaccines deliver transgenic mRNA through lipid nanoparticles as carriers. Viral vector vaccines utilize adenovirus and integrate genetic material from SARS-CoV-2 into its genome. Inactivated vaccines involve SARS-CoV-2 that has been killed by physical or chemical means.

**Figure 2 vaccines-10-00742-f002:**
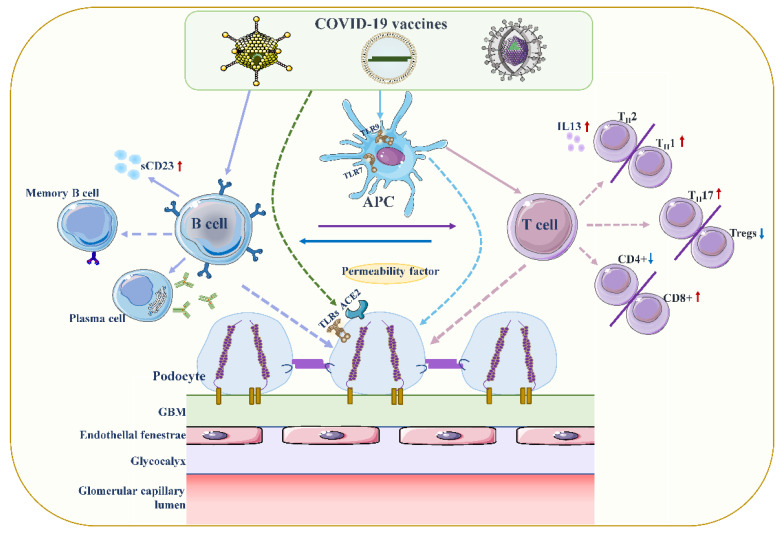
Proposed mechanisms of podocyte injury caused by COVID19 vaccination. Vaccination stimulates antigen-presenting cells (APCs) and B cells, which in turn activate T cells through antigen presentation and cytokine production. A decrease in CD4+ T helper (Th) cells is associated with the prevalence of CD8+ cytotoxic T cells, and an imbalance between Th2 and Th1 cells is associated with an increase in Th2-specific interleukin-13 (IL-13) production, and Th17. In contrast to increased cellular activity, the frequency and function of regulatory T cells (Tregs) decreased. Permeability proteins, such as cytokines and autoantibodies, can directly affect podocytes, leading to loss of foot processes and disruption of the glomerular permeability barrier. In addition, the vaccine can also affect podocytes through specific toll-like receptors (TLRs), and angiotensin conversion enzyme 2 (ACE2). The figure refers to the pathogenesis of minimal change disease by Vivarelli et al [53].

**Table 1 vaccines-10-00742-t001:** Summary of published cases of newly diagnosed acute kidney disease.

Case	Authors	Age/Sex	Country (Race)	Medical History	Vaccine			Onset (Day)	Baseline-Scr (mg/dL)	After Vaccine-Scr (mg/dL) (Day)	Newly HT/Worse	Symptoms	Diagnosis	Treatments	Outcomes
					Type	Manufacturer	Onset after Which Dose								
New Case															
1	Leclerc et al. [16]	71/M	Canada	dyslipidemia treated with rosuvastatin	Vector	AstraZeneca	1st	D1	0.7	10.6 (D14)	III	edema	MCD	HD, mPSL 1 g/day 1–3 day, PSL 60 mg/day	CR. Scr was 1.2 mg/dL, UPCR 28 mg/mmol at D81
2	Lim et al. [17]	51/M	Korea	None	Vector	Janssen	1st	D7	NA	1.54 (D28)	I	edema	MCD	mPSL 64 mg/day	CR. Scr was 0.95 mg/dL, UPCR was 0.2 g/g at D57
3	Lebedev et al. [18]	50/M	Israel	None	mRNA	Pfizer-BioNTech	1st	D4	0.78	2.31 (D10)	III	edema, abdominal pain, diarrhea	MCD with ATI	PSL 80 mg/day	CR. Scr was 0.97 mg/dL, UACR was 155 mg/g at D37
4	Maas et al. [19]	80s/M	Netherlands	VTE	mRNA	Pfizer-BioNTech	1st	D7	NA	1.43 (D7)	II	edema	MCD with ATI	PSL 80 mg/day	CR. UPCR was 0.68 g/g after 10 days of PSL
5	D’Agati et al. [20]	77/M	USA (Caucasian)	T2DM	mRNA	Pfizer-BioNTech	1st	D7	1.0–1.3	2.33 (D14)	I	edema	MCD with ATI	mPSL 1 g/day 1–3 day, PSL 60 mg/day	NR. Scr was 3.74 mg/dL, UTP was 18.8 g/day at D35
6	Holzworth et al. [21]	63/F	USA	HT	mRNA	Moderna	1st	<D7	0.7	1.48 (>D28)	III	edema, dyspnea	MCD with ATI and AIN	mPSL 500 mg/day 1–3 day, PSL 1 mg/kg/day	NA
7	Weijers et al. [22]	61/F	Netherlands	AIH, hypothyroidism	mRNA	Pfizer-BioNTech	1st	D1	0.7–0.8	1.47 (D4)	NA	edema	MCD	HD, steroids 1 mg/kg/day	CR. Scr was <1 mg/dL at D77, UTP was 0 g/day at D58
8	Kobayashi et al. [23]	75/M	Japan	edema and hydrocele testicle after 1st vaccine	mRNA	Pfizer-BioNTech	2nd	D2	0.96	1.24 (D7)	I	edema	MCD	mPSL 1 g/day 1–3 day, PSL 1 mg/kg/day	CR was achieved within D42
9	Lim et al. [52]	51/M	Korea	None	Vector	Janssen	1st	D7	Normal	1.54 (D21) 1.96 (D33)	NA	edema	MCD	high-dose steroid	CR was achieved after 3 weeks of treatment
10	Hanna et al. [24]	60/M	Canada	None	mRNA	Pfizer-BioNTech	1st	D10	0.89	1.34 (D45)	II	edema, dyspnea	MCD with ATI	PSL 80 mg/day	R. Scr was 1.03 mg/dL at 11 weeks
11	Klomjit et al. [11]	83/M	USA (Caucasian)	NA	mRNA	Moderna	2nd	D28	1.19	2.19	NA	AKI	MCD, ATN	high-dose steroid	R. Scr was 1.2 mg/dL during last follow-up
12	Da et al. [26]	70/M	Singapore	edema after 1st vaccine	mRNA	Pfizer-BioNTech	2nd	D1	NA	1.28	I	edema	MN (anti-PLA2R-)	irbesartan, frusemide, warfarin	NR within D60
13	Gueguen et al. [27]	76/M	France	HT, UV-treated cutaneous mycosis fungoid	mRNA	Pfizer-BioNTech	1st	D4	0.86	1.14	NA	edema	MN (anti-PLA2R 1:800)	RASB	PR.
					mRNA	Moderna	2nd	D2	1.14	1.15	NA	edema	MN	RTX 1 g 1–14 day	PR.
14	Kudose et al. [28]	50/F	USA (Caucasian)	HT, obesity, APS	mRNA	Moderna	2nd	D2	1.3	1.7	NA	gross hematuria, fever, body aches	IgAN (M1E0S1T1C1)	conservative	CR. hematuria resolved within D5
15	Kudose et al. [28]	19/M	USA (Caucasian)	microhematuria	mRNA	Moderna	2nd	D2	Normal	1.2	NA	gross hematuria	IgAN (M1E1S1T0C0)	conservative	CR. hematuria resolved within D2
16	Tan et al. [29]	41/F	Chinese	GDM	mRNA	Pfizer-BioNTech	2nd	D1	Normal	1.73 (D2)	I	gross hematuria, headache, myalgia	IgAN with fibrocellular and fibrous crescents	pulse mPSL, PSL, CyC	NA
17	Hanna et al. [30]	17/M	USA (Caucasian)	foamy urine	mRNA	Pfizer-BioNTech	2nd	<D1	Normal	1.78 (D6)	I	gross hematuria	IgAN (M1E1S1T1C1)	pulse mPSL	R. Scr improved (duration not reported)
18	Anderegg et al. [31]	39/M	Switzerland	HT	mRNA	Moderna	2nd	immediately	NA	AKI	NA	flu-like symptoms, fever, macrohematuria	severe crescentic IgAN	high-dose glucocorticoids, CyC	R. Scr was normalized within several weeks
19	Klomjit et al. [11]	38/M	USA (Caucasian)	NA	mRNA	Pfizer-BioNTech	2nd	D14	1.3	1.6	NA	gross hematuria	IgAN	conservative	NA
20	Klomjit et al. [11]	44/M	USA (Caucasian)	NA	mRNA	Moderna	1st	D14	1.1	2.5	NA	AKI	IgAN, AIN	high-dose steroid	NR. Scr was 3.6 mg/dL during last follow-up
21	Klomjit et al. [11]	66/M	USA (Caucasian)	NA	mRNA	Moderna	1st	D14	1.1	1.5	NA	gross hematuria	IgAN	PSL	R. Scr was 1.4 mg/dL during last follow-up
22	Klomjit et al. [11]	62/M	USA (Caucasian)	NA	mRNA	Pfizer-BioNTech	2nd	D42	1.0	2.2	NA	AKI	IgAN	conservative	R. Scr was 2.0 mg/dL during last follow-up
23	Tan et al. [29]	60/F	Malay	hyperlipidemia	mRNA	Pfizer-BioNTech	2nd	D1	Normal	6.11 (D39)	III	gross hematuria	Anti-GBM nephritis	pulse mPSL, PSL, CyC, PLEX	NA
24	Sacke et al. [32]	older/F	USA	None	mRNA	Moderna	2nd	D14	Normal	7.8	NA	fever, gross hematuria, anorexia, nausea	Anti-GBM with mesangial IgA deposits	mPSL, CyC, PLEX	NR. remained HD dependent
25	Klomjit et al. [11]	77/M	USA (Caucasian)	NA	mRNA	Pfizer-BioNTech	1st	D7	1	1.8	+	HT	Atypical anti-GBM nephritis	PSL, mycophenolate	NR. Scr was 2.9 mg/dL during last follow-up
26	Sekar et al. [35]	52/M	USA (Caucasian)	HT	mRNA	Moderna	2nd	D1	1.11	8.41 (D14)	NA	headache, weakness	PR3-ANCA vasculitis	RTX, CyC, PSL, HD	NR. remained HD dependent
27	Anderegg et al. [31]	81/M	Switzerland	sustained flu-like symptoms after 1st vaccine	mRNA	Moderna	2nd	<D1	NA	AKI	NA	flu-like symptoms worsened	PR3-ANCA vasculitis	high-dose glucocorticoids, CyC, PLEX	R. renal function improved within D21
28	Feghali et al. [38]	58/M	USA (Caucasian)	None	mRNA	Moderna	2nd	D4	NA	4.1	NA	hematuria, proteinuria	PR3-ANCA vasculitis	mPSL 1 g 1–3 day, PSL 60 mg/kg/day, RTX, CyC, PLEX	R. Scr was 1.5 mg/dL after 10 weeks of diagnosis
29	Villa et al. [33]	63/M	Spain	None	Vector	AstraZeneca	1st	D2	Normal	2.9 (D7)	NA	flu-like syndrome, hemoptysis	MPO-ANCA vasculitis	high-dose glucocorticoids, CyC	NR. Scr was 2.08 mg/dL at D49
30	Hakroush et al. [34]	79/F	Italy (Caucasian)	HT, degenerative disc disease	mRNA	Pfizer-BioNTech	2nd	D14	0.71	1.38 (D14) 6.57 (D24)	NA	weakness, upper thigh pain	MPO-ANCA vasculitis, ATI	mPSL 250 mg/day 1–3 day, PSL 1 mg/kg/day, CyC	R. Scr was normalized within D47
31	Klomjit et al. [11]	82/F	USA (Caucasian)	NA	mRNA	Moderna	2nd	D28	0.8	2.5	NA	AKI, hematuria, proteinuria	MPO-ANCA vasculitis	High-dose steroid, RTX	R. Scr was 2.3 mg/dL during last follow-up
32	Shakoor et al. [36]	78/F	USA	T2DM, HT, atrial fibrillation	mRNA	Pfizer-BioNTech	1st	<D7	0.77	1.31 (D16)	NA	nausea, vomiting, diarrhea	AKI	None	CR. improved spontaneously
					mRNA	Pfizer-BioNTech	2nd	D6	Normal	3.54 (D6)	NA	lethargy, nausea, vomiting, diarrhea	MPO-ANCA vasculitis	mPSL 1–3 day, PSL 1 mg/kg/day, RTX	R. Scr was 1.71 mg/dL at 1-month follow-up
33	Dube et al. [37]	29/F	USA	congenital diffuse cystic lung disease	mRNA	Pfizer-BioNTech	2nd	D16	0.8	1.25 (D16) 1.91 (D49)	Normal	NA	MPO-ANCA vasculitis	mPSL 500 mg 1–3 day, PSL 1 mg/kg/day, RTX, CyC	R. Scr was 1.01 mg/dL at D133
34	Gillion et al. [39]	77/M	Belgium	None	Vector	AstraZeneca	1st	D28	1.2	2.7	NA	fever, night sweat	ANCA-negative granulomatous vasculitis	mPSL	R. Scr was normalized within D56
35	Mira et al. [40]	45/F	Portugal (Caucasian)	total thyroidectomy	mRNA	Pfizer-BioNTech	2nd	D1	0.85	18.4 (D8)	Normal	anorexia, nausea, vomiting, urine output reduction	AIN, ATI	HD, mPSL 500 mg/day 1–3 day, PSL 1 mg/kg/day	R. Scr was 1.02 mg/dL at D37
36	Unver et al. [25]	67/F	Turkey	T2DM, MCD in PR	Inactivated	Sinovac Life Science	2nd	D10	0.8	4.2 (D26)	III	edema, headache	AIN, ATI	mPSL 500 mg/day 1–3 day, PSL 1 mg/kg/day, cyclosporine A	PR. Scr was 1.12 mg/dL at D60, UTP was 3 g/day at D115
37	Lim et al. [52]	44/M	Korea	T2DM, chronic hepatitis B infection, hyperlipidemia	mRNA	Moderna	1st	D1	0.91	4.13 (D7) 4.94 (D21)	NA	gastrointestinal discomfort, anorexia	ATN	high-dose steroid	PR. Scr was 1.89 mg/dL, UPCR was 0.3 g/g at D42
38	Lim et al. [52]	77/F	Korea	T2DM, Chronic hepatitis B, hepatocellular carcinoma	mRNA	Pfizer-BioNTech	2nd	D1	0.98	10.67 (D7) 11.15 (D14)	NA	severe nausea and vomiting	ATN with myoglobin tubular casts	HD	PR. Scr was 2.12 mg/dL, within 4 months
39	Missoum et al. [50]	58/M	Algeria	HT	Inactivated	Sinovac Life Science	1st	D9	Normal	8.9	NA	fever, arthralgias, purpura	Leukocytoclastic vasculitis ATN	HD, prednisone	R. Scr was 2.8 mg/dL at D90
40	Al Rawahi et al. [51]	64/M	Sultanate of Oman	HT, hyperlipidemia	Vector	AstraZeneca	1st	D7	NA	1.18 (D7)	I	fever, lethargy, abdominal pain	aTTP, VITT	argatroban, fondaparinux, hydrocortisone, immunoglobulin	R. renal function improved at D15
41	Yocum et al. [44]	62/F	USA	hyperlipidemia, GERD, hypothyroidism, HT	Vector	Janssen	1st	D37	NA	2.19 (D37) 6 (D38)	III	altered mental status	aTTP, VITT,	PLEX, HD, mPSL, packed RBCs	NA
42	Osmanodja et al. [46]	25/M	Germany	None	mRNA	Moderna	1st	D2	NA	1.5 (D13)	NA	fever, headache, petechiae	aTTP	PLEX, PSL 250 mg 1–3 day, caplacizumab	R. Scr was 1 mg/dL at D27
43	Alislambouli et al. [48]	61/M	Korean-American	NA	mRNA	Pfizer-BioNTech	1st	D5	NA	1.57 (D5)	NA	fever, confusion, headache, emesis, ecchymosis	aTTP	PLEX, mPSL 1 g 1–3 day, RTX	R. rapid and excellent response
44	Yoshida et al. [49]	57/M	Japan	None	mRNA	Pfizer-BioNTech	1st	D7	NA	1.57 (D14)	NA	fatigue, loss of appetite, jaundice	aTTP	PLEX, PSL, RTX	R. in good condition at D48
45	Ruhe et al. [45]	84/F	Germany	NA	mRNA	Pfizer-BioNTech	1st	D16	NA	1.95 (D16)	III	partial hemiplegia, petechiae	aTTP	PLEX, RTX, corticosteroid	R. Scr was 0.6 mg/dL at D34
46	Chamarti et al. [47]	80/M	Hispanic	HT, T2DM, hyperlipidemia, gout, IDA	mRNA	Pfizer-BioNTech	2nd	D12	NA	2.4 (D14)	I	generalized weakness, malaise	aTTP	PLEX, packed RBCs, platelets, prednisone	R. Scr was 1 mg/dL at D30
47	Lim et al. [52]	69/F	Korea	T2DM	Vector	AstraZeneca	1st	D2	0.8	3.69 (D14)		general weakness, gastrointestinal discomfort	aTTP	None	CR Scr was 0.65 mg/dL, UPCR was 1.0 g/g at D56
Relapsed cases															
48	Mancianti et al. [41]	39/M	Italy (Caucasian)	MCD in remission for 37 years	mRNA	Pfizer-BioNTech	1st	D3	0.9	1.8 (D8)	NA	edema	MCD	PSL 1 mg/kg/day	CR
49	Klomjit et al. [11]	67/F	USA (Caucasian)	MCD	mRNA	Moderna	2nd	D21	1	1.6	NA	edema	MCD	high-dose steroid, RTX	R. Scr was 1.5 mg/dL, UTP was 0.07 g/day during last follow-up
50	Aydin et al. [42]	66/F	Turkey	hyperlipidemia, DM, HT, MN in CR for 8 years	Inactivated	Sinovac Life Science	1st	D14	Normal	2.78 (D14)	NA	edema	MN (anti-PLA2R 1:120.53)	NA	NA
51	Klomjit et al. [11]	39/M	USA (Caucasian)	MN	mRNA	Pfizer-BioNTech	2nd	D7	0.91	1.13	NA	edema	MN (anti-PLA2R+)	TAC	R. Scr was 1.1 mg/dL, UTP was 5.7 g/day during last follow-up
52	Hanna et al. [30]	13/M	USA (Caucasian)	IgAN, T1DM	mRNA	Pfizer-BioNTech	2nd	<D1	0.54	1.31 (D2)	NA	gross hematuria, vomiting	IgAN (M0E0S0T0C0)	conservative	CR. hematuria and Scr resolved within D6
53	Perrin et al. [43]	41/F	France	IgAN, KT	mRNA	Pfizer-BioNTech	1st	D2	NA	Scr transiently increased	NA	gross hematuria	IgAN	conservative	CR. spontaneously resolved

Abbreviations: AIH, autoimmune hepatitis; AIN, acute interstitial nephritis; ATN, acute tubular necrosis; ATI, acute tubular injury; APS, antiphospholipid syndrome; ANCA, anti-neutrophil cytoplasmic autoantibodies; ANCA GN, anti-neutrophil cytoplasmic autoantibody-associated glomerulonephritis; anti-PLA2R, anti-phospholipase A2 receptor; aTTP, acquired thrombotic thrombocytopenic purpura; CyC, cyclophosphamide; CR, complete remission; CRF, chronic renal failure; DM, diabetes mellitus; F, female; FSGS, focal segmental glomerulosclerosis; GBM, glomerular basement membrane; GDM, gestational diabetes; GERD, gastroesophageal reflux disease; HD, hemodialysis; HT, hypertension; IDA, iron deficiency anemia; IgAN, IgA nephropathy; ITT, vaccine-immune thrombotic thrombocytopenia; KT, kidney transplantation; M, male; MCD, minimal change disease; mPSL, methylprednisolone; MN, membranous nephropathy; MPA, mycophenolic acid; NA, not applicable; NR, no response; PR, partial remission; PLEX, plasma exchange; PSL, prednisolone; R, response; RASB, renin-angiotensin system blockade; RBCs, red blood cells; RTX, rituximab; Scr, serum creatinine; TAC, tacrolimus; T1DM, Type 1 diabetes mellitus; T2DM, Type 2 diabetes mellitus; UACR, urinary albumin-creatinine ratio; UTP, 24-h urine protein; UPCR, urine protein-to-creatinine ratio; VITT, vaccine-induced immune thrombotic thrombocytopenia; VTE, venous thromboembolism.

## Data Availability

Not applicable.

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
