# Peer review of "New-Onset Acute Kidney Disease Post COVID-19 Vaccination"

_vaccines, 2022, doi:10.3390/vaccines10050742_

Round 1

Reviewer 1 Report

I commend the authors for this comprehensive review summarising clinical features of AKD after vaccination and possible mechanisms of AKD after COVID-19 vaccination. I have provided a few suggestions below to consider:

  1. Please provide an example for based vaccines as it is provided for others in line 30
  2. The Author may like to revisit the title for Figure 1. "SARS-CoV-2 and the main types of vaccines for AKD after vaccination."
  3. Please confirm the case number in Table 1, numbering seems to be distorted
  4. When mentioning about searching PubMed, it will be great if the author lists a few of the keywords that the author used while searching. Also please mention what was the timeline of when these searches were conducted. I understand this is not a systematic review; however, providing this additional information will further improve the credibility of the paper. 
  5. Furthermore, there is a recent case series reporting new-Onset Kidney Diseases after COVID-19 Vaccination published Vaccines (Basel). 2022;10(2):302. The author may like to include this.
  6. Also, were only case reports selected, or were there only case reports found after the PubMed search?
  7. I suggest at least for the title of the sections such as 3.1 Please use the full form instead of the abbreviation. I understand the author has already expanded these abbreviations in the first occurrence; however, the heading can still be put in full form. Like section 4.3.
  8. In the end, the author should mention at an appropriate place (likely in the discussion section) something that says, "Despite these reported cases of AKD, the protective benefits of COVID-19 vaccination far outweigh any risks identified so far." 

Reviewer 2 Report

 Although COVID-19 vaccines have shown to be effective in providing protection against SARS-CoV2 infection and reducing severe disease progression even after the virus infection, adverse effects caused by immunization after current COVID-19 vaccines are safety concerns.  The authors attempt to provide a review of 43 case reports of individuals who developed acute kidney diseases after vaccinations, which would provide valuable guidance for clinicians to provide care for vaccinees with unforeseen adverse effects and insights for the improvement of vaccine development. Despite the topic being highly relevant and important, the authors did not deliver the key message clearly.

I was lost while I was reading the subsections 4.2 through 4.4. There are many parts where English is a concern because some parts are vague and confusing.  I would suggest that the authors revise the current manuscript possibly with help from colleagues or collaborators whose native language is English.

Overall, it is not clear what the take-home messages of this review are.  The current form of the review seems to be premature to extract meaningful insights.  It may be helpful to run a demographical analysis of the 43 cases of COVID-19 vaccination based on the age, gender, and vaccine types (mRNA-based vaccines vs. non-mRNA vaccines) of the patients who developed AKD after the COVID-19 vaccination.

Specific comments are listed below.

Lines 55-56, ‘relapse acute kidney disease (AKI)’ does not match the acronym. Do the authors mean ‘acute kidney injury (AKI)’?  The authors need to define the acronyms clearly between ‘AKI’ and ‘AKD’.

Line 59, be consistent in the description of AKI stage; maybe use “1.5 to 1.9 times increase” instead of “1.5- to 1.9-fold increase” to be consistent with the rest of the description.

Line 78, change ‘NA, nonapplicable’ to ‘NA, not applicable'

There are multiple places where the author used numeric numbers at the beginning of the sentence, which need to be spelled out. For example, 

Line 104, change from ‘6 patients...’ to ‘Six patients’ and from ‘9 patients...’ to ‘Nine patients’,

Line 109, change from ’10 patients...’ to ‘Ten patients’,

Line 113, change from ’11 cases..’ to ‘Eleven cases’,

Line 115, change from ‘6 patients...’ to ‘Six patients’

Line 135, change from ‘2 patients...’ to ‘Two patients’

Line 144, change from ‘1 case...’ to ‘One case’

Check throughout the manuscript and change these whenever applicable.

Line 159-160, consider revising the sentence.

Line 190, a period ‘.’ is missing ‘… Th17 In contrast...’ 

Lines 201- 203, the sentence ‘..self-NA release…. caused by podocyte disease [23]’ is not clear.

  • Particularly, I am not following ‘..and via Th2 Allergic inflammation caused by podocyte disease.’ It does not make sense.  Please clarify.
  • Change from ‘Th2 Allergic inflammation’ to ‘Th2-mediated allergic inflammation’

Lines 213-219 need a revision for clarification

  • Lines 213-215, delete ‘autoantibodies’ after ANCA. Because according to the authors’ definition, ANCA stands for anti-neutrophil cytoplasmic antibodies, which contain the meaning of being autoantibodies by themselves. Also, consider defining the ANCA as ‘anti-neutrophil cytoplasmic autoantibodies’.
  • Lines 216-217, this sentence contradicts the previous sentence. Do the authors mean that ‘ANCA response triggered by RNA vaccines was significantly reduced after the treatment of vaccinees with ribonuclease’?
  • Lines 217-219, ‘ANCA-associated vasculitis (AAV) and autoimmune responses imply a direct response to viral RNA’ Does the author mean ‘it implies that a host response to viral RNA can directly cause AAV and autoimmune response’? If yes, what’s written doesn’t mean this. It needs a clarification.
  • Line 219, change ‘COVID19’ to ‘COVID-19’ to be consistent

Line 225, delete ‘in’ from ‘.. viral antigens in and promoting..’

Line 227, correct from ‘in’ to ‘by’ or revise the sentence

Lines 229-231 do not make sense. Revise for clarification. 

For example,

  • A subject is missing.
  • Change from ‘mRNA’ in the beginning of the sentence to ‘Messenger RNA’
  • Also, the message is not clear

 Lines 241-243, revise for clarity

Lines 246-247, the statement ‘the virus in the vaccine is not copyable’ is incorrect.  mRNA vaccine is not a full-length viral genome. So, there is “no virus in vaccine”. Does the author mean’ mRNA vaccine is replication-incomparable’?

Lines 247-249, ‘ .. digestive system is not too possible to lead to thrombosis…’ is not clear. Please revise.

 Lines 257 -259, the sentence is not clear. Revise.

Line 268, the statement ‘AKD is a rare but uncommon diseases’ is contradicting, needs to revise. 

Lines 289- 290, it is weird that ‘acute kidney disease’ was represented by two acronyms. Clarify what “I” stands for on line 290.

Round 2

Reviewer 2 Report

The authors' responses to the specific comments and revision were satisfactory and improved for clarification.